

# Lost, hidden, broken, cut-estimating and interpreting the shapes and masses of damaged assemblages of plesiosaur gastroliths

Donald M. Henderson

Royal Tyrrell Museum of Palaeontology, Drumheller, Alberta, Canada

## ABSTRACT

**Background:** Gastroliths are stones of uncertain purpose that are commonly found inside the rib cages of plesiosaur fossils worldwide. Gastroliths from four Alberta (Canada) plesiosaurs were studied to determine both their shapes and masses, and their mass fractions relative to body mass. One animal's set of gastroliths was 100% complete and fully visible, but the others showed varying degrees of loss, damage or obscuration, so estimations of their original states were needed.

**Methods:** The studied animals were: *Albertonectes vanderveldei*, *Fluvionectes sloanae*, *Nichollssaura borealis* and *Wapuskanectes betsynichollsae*. The animals come from three different palaeoenvironments: open marine, near shore marine, and fluvial. Gastrolith shapes were classified as either xiphoid, cylindrical, discoidal or spherical based on observed and/or estimated dimensions. Although not all methods could be applied in all cases, gastrolith shapes and masses were estimated four different ways: (1) direct measurement and weighing of a subset and predicting the properties of the remaining obscured and hidden stones; (2) measuring triaxial ellipsoid dimensions of free stones to calculate volumes and multiplying by the mass density of chert; (3) measuring two visible triaxial dimensions of embedded stones, estimating the hidden third dimension three different ways, and then determining volumes and masses by calculation; and (4) predicting the density and mass of a densely packed cluster of small gastroliths using geometrical arguments.

**Results:** Total gastrolith mass never exceeded 0.2% of body mass in any plesiosaur, and is consistent with the idea that the amounts of gastroliths recovered with plesiosaurs would be ineffective as ballast. The largest plesiosaur in the sample had the largest single gastrolith and total gastrolith mass increases with body size. The shape characteristics of the gastroliths were different for different environments, but compositionally they are dominated by black cherts. A possible common source for the gastroliths was identified for the two geographically close and near-contemporanous *Nichollssaura* and *Wapuskanectes*.

Corresponding author
Donald M. Henderson,
don.henderson@gov.ab.ca

## INTRODUCTION

Plesiosaurs were unusual animals with nothing like them living today which makes them both very interesting and very frustrating animals to study and understand. The last of the plesiosaurs went extinct at the end of the Cretaceous (*Benton, 1997*), and their evolutionary origins are obscure, but they are diapsid reptiles and may be more closely related to archosaurs than to lizards and snakes (*Bardet et al., 2023*). They were among the first vertebrate fossils to be studied in a rigorous, scientific fashion beginning in the early 19th century (*De la Beche & Conybeare, 1821*; *Ellis, 2003*), and gastroliths (colloquially known as "stomach stones") are frequently found with their body fossils. The purpose of the stones has been controversial ever since they were first encountered, with the most common suggestions being that they were either for digestion, ballast, or material accidentally ingested. See *Taylor (1993)* and *Wings (2007)* for comprehensive reviews.

Several relatively complete plesiosaurs have been discovered in the western Canadian province of Alberta over the past 30 years, with the most recent one being from December 2012–not counting more fragmentary specimens from 2019 and 2023. However, not all of these recent discoveries have been formally described. These fossils come from two different time periods–Early Cretaceous (Aptian-Albian boundary) and Late Cretaceous (late Campanian)–and from three very different environments-open marine, near-shore marine, and fluvial. Within the intact rib cages of some of these specimens have been found collections of gastroliths, and it was realized that a quantitative analysis of the stomach stones and the bodies that hosted them could be done with the knowledge that the stones really did come from within the specimens and were not washed into the body after death. The rock units hosting the fossils are devoid of any gravels, pebbles or cobbles, demonstrating that the gastroliths within the bodies were genuinely allochthonous. However, the hosting fossils were either subject to varying degrees of erosion prior to their discovery, or were damaged when first exposed, leading to a loss of fossil material and often a partial loss of any enclosed gastroliths. Additionally, in all but one case (see below), decisions about the preparation of the fossils resulted in bones and any enclosed gastroliths not being removed, thus leaving quantities of stones incompletely exposed or fully hidden inside the bodies.

Despite uncertainty about their actual biological function(s), any new analyses of plesiosaur gastroliths and their relationship to other aspects of the bodies of the animals have the potential to improve our understanding of these intriguing geological-biological entities. This article presents the results of using quantitative methods to make estimates of the original masses and shape characteristics of incompletely preserved, broken or obscured stomach stones found with plesiosaur fossils. Lastly, the body masses of these plesiosaurs, derived from three-dimensional, digital models of the animals, are used to compare the relative mass fractions of the gastroliths to the bodies that hosted them.

## MATERIALS

### Body fossils

The geographic locations of the four species are shown in Fig. 1A. Three of the studied specimens are based on single individuals: *Albertonectes vanderveldei* TMP2007.11.1,

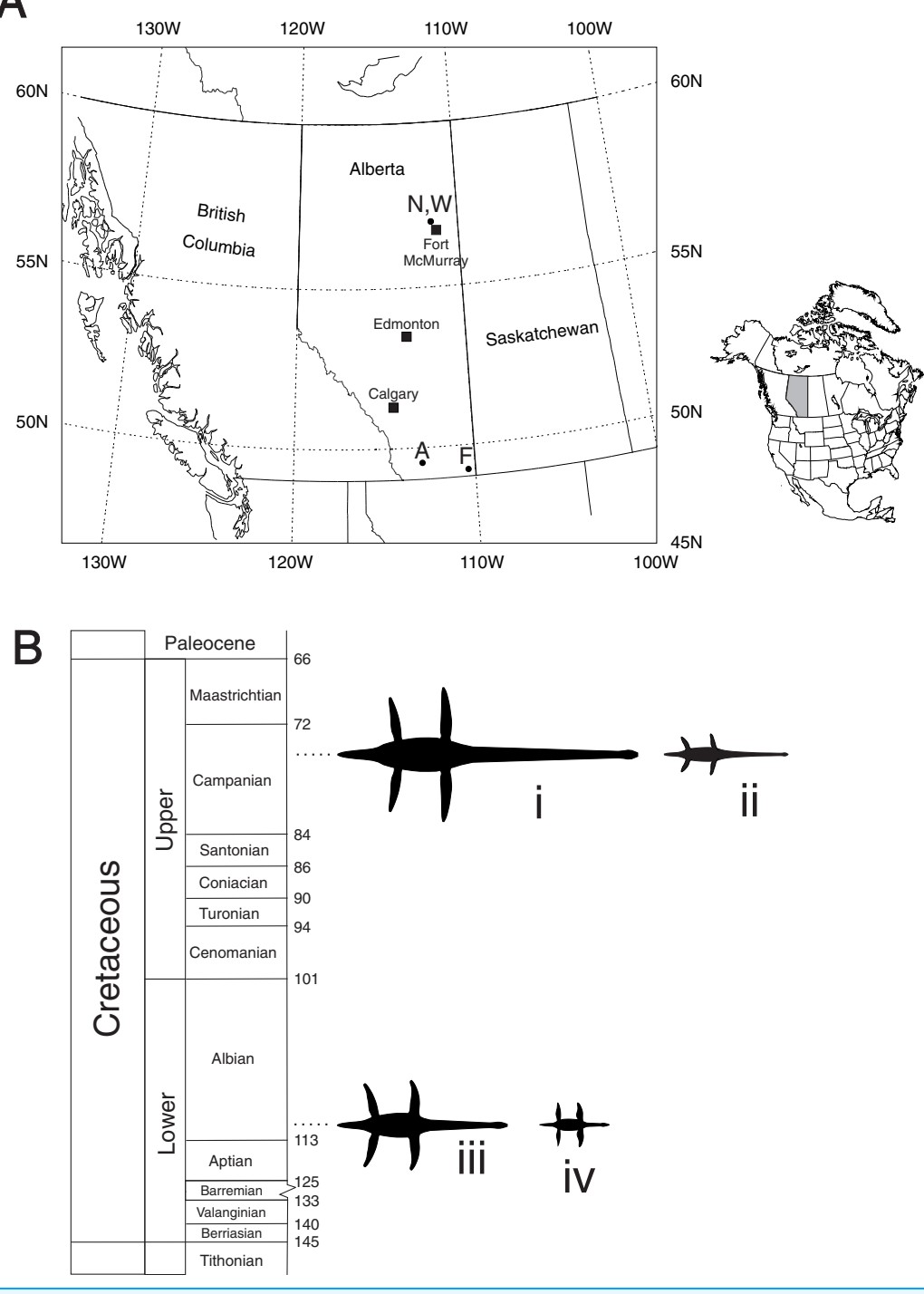

**Figure 1 Geographic and stratigraphic locations of the four Cretaceous plesiosaurs from the province of Alberta, Canada discussed in the text.** (A) Location of the province of Alberta in western Canada and North America. Specimen locations are labelled as follows—A: *Albertonectes vanderveldei* TMP2007.11.1, Bearpaw Formation, late Campanian. F: *Fluvionectes sloanae*, TMP2009.37.68, Dinosaur Park Formation, late Campanian. N: *Nichollssaura borealis* TMP1994.122.1, Clearwater Formation, early Albian. W: *Wapuskanectes betsynichollsae* TMP2011.88.1 and TMP2012.50.1, Clearwater Formation, early Albian. (B) Stratigraphic positions of the four plesiosaurs. *i. Albertonectes vanderveldei ii. Fluvionectes sloanae. iii. Wapuskanectes betsynichollsae iv. Nichollssaura borealis* Dorsal silhouettes are

**Figure 1 (continued)**
to scale with *A. vanderveldei* at 12 m. Chronostratigraphic chart after *Cohen et al. (2013)*, with the numbers on the righthand side indicating boundary ages in millions of years. Silhouettes derived from the 3D digital models of the animals that were specially created for this study by the author (See Fig. 7). Maps produced by the author using the mapping software provided with the scientific visualization software PV-WAVE (Perforce Software, Inc.).                               

*Fluvionectes sloanae* TMP2009.37.68, *Nichollssaura borealis* TMP1994.122.1; while the analysis of *Wapuskanectes betsynichollsae* is based on two separate specimens—TMP2011.88.1 and TMP2012.50.1. Three of the specimens were found during open pit mining—*Albertonectes*, *Nichollsaura*, and *Wapuskanectes* TMP2011.88.1. Another specimen of *Wapuskanectes*, TMP2012.55.1, was exposed during roadworks, while *Fluvionectes* was discovered during field prospecting. Figure 1B shows the chronostratigraphic positions of the four genera.

    *Albertonectes* (Fig. 2) was found in 2007 during open pit mining for gem quality ammonite shells ("ammolite") south of the town of Lethbridge in the mine operated by Korite International. It comes from the Late Cretaceous Bearpaw Formation and was described and named as the elasmosaurid *Albertonectes vanderveldei* by *Kubo, Mitchell & Henderson (2012)*. This specimen is missing its head, its left forelimb and portions of the distal regions of the hind limbs, and these losses happened prior to final burial. However, all the vertebrae from the atlas-axis to the tip of the tail, and most of the ribs, were found in articulation, although the neck had snapped into four sections. Gastroliths with this specimen were first observed in the field, with more being revealed during preparation.

    *Fluvionectes* (Fig. 3) was found during fossil prospecting in the extreme southeast of the province in 2009 in an area known as the Porcupine Pocket of the Sage Creek Grazing Preserve. It is another elasmosaurid (*Campbell et al., 2021*) and was found in rocks of the Lethbridge Coal Zone, a subdivision of the Dinosaur Park Formation (DPF), and the Zone represents a near-coastal setting that was subject to tidal influences (*Eberth, 2005*). The biozone defined by the ammonite *Baculites compressus* is the same for the Bearpaw Formation that produced the *Albertonectes* and the Lethbridge Coal Zone (*Tsujita, 1995*). *Fluvionectes* was found in fluvial sediments along with many pieces of coalified wood in close association with the bones, and the interpretation is that the carcass was caught in a log jam (*Campbell et al., 2021*). The periphery of the outcrop which held the specimen was heavily eroded when found, and much of the animal's extremities were lost. This included most of the tail, the neck and head, and all the proximal half of the left forelimb. However, most of the axial body from the proximal quarter of the tail to the proximal quarter of the neck is preserved, with all the bones in close association, and this allows an estimation of the animal's size to enable a body reconstruction. The presence of gastroliths with this specimen was revealed during preparation.

    *Nichollssaura* (Fig. 4) was uncovered during overburden removal in the Syncrude Canada Ltd. tar sand mine north of Fort McMurray, and comes from the Wabiskaw Member of the Early Cretaceous Clearwater Formation which is interpreted to be earliest Albian in age (*Caldwell et al., 1978*). This specimen was almost complete except for a

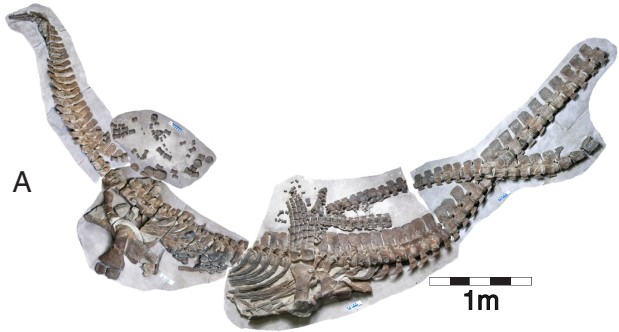

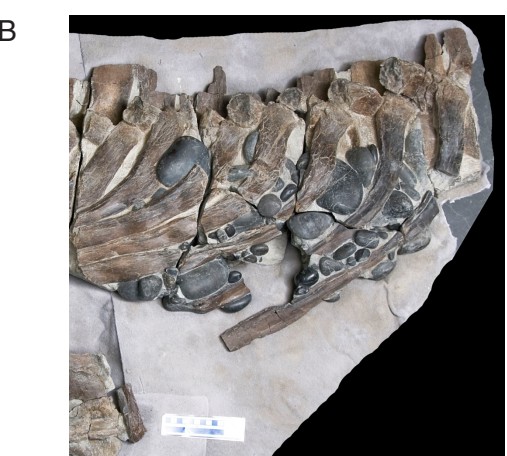

**Figure 2** *Albertonectes vanderveldei* TMP2007.11.1 (*Kubo, Mitchell & Henderson, 2012*). (A) View of the original downside in the field and the best preserved side as it experienced the least disruption by scavenging sharks. (B) Detail of midbody region showing the large gastroliths that settled downwards onto the inside of the right, dorsal ribs. This region was hit by the excavator bucket with the partial loss of ribs and an unknown number of gastroliths. Fortuitously, the broken blocks hosting gastroliths can be removed and their contents inspected from several sides. Scalebar is 10 cm. Source credit: Sue Sabrowski, Royal Tyrrell Museum of Palaeontology. 

missing left forelimb, the bones distal to the ankle on the right hindlimb, and the proximal third of the cervical ribs on the left side, and the first few anterior dorsal ribs, also on the left side. These missing elements were reconstructed for display purposes. This animal is a leptocleidid plesiosaurian and was described by *Druckenmiller & Russell (2008)*. The presence of its gastroliths was revealed during preparation.

*Wapuskanectes* specimen TMP2011.88.1 (Fig. 5) is an early elasmosaurid and comes from the same Syncrude Canada Ltd. tar sand mine and rock unit that produced *Nichollssaura*–the Wabiskaw member of the Clearwater Formation. The genus was described and named by *Druckenmiller & Russell (2006)*, with the holotype specimen being TMP1998.49.2 which also came from the same mine and rock unit. The present specimen consists of proximal neck vertebrae, anterior dorsal vertebrae, the pectoral region, a left humerus, some manual phalanges, and some incomplete ribs and gastralia. This specimen has a substantial collection of gastroliths visible at the posterior end of the pectoral girdle that was revealed during preparation.

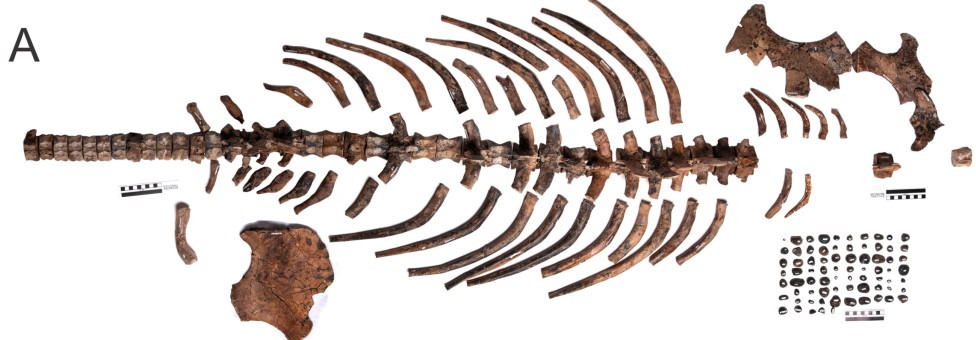

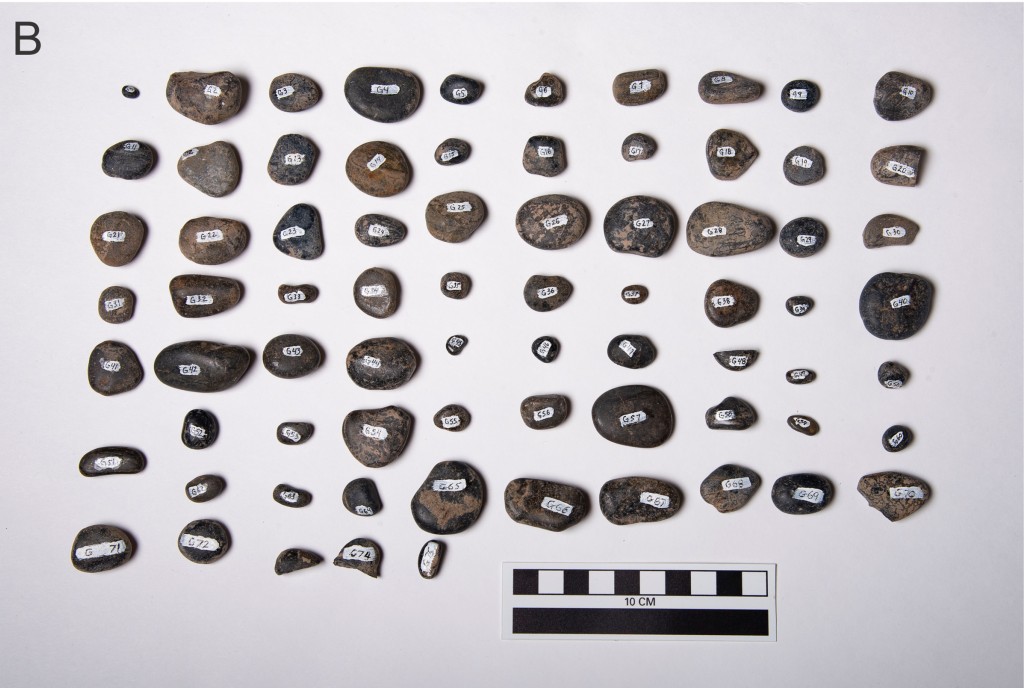

**Figure 3** *Fluvionectes sloanae* TMP2009.37.68 (*Campbell et al., 2021*). (A) Recovered skeletal and gastrolith material with the gastroliths arranged in a grid at the lower right. (B) Full complement of gastroliths removed from inside the rib cage of TMP2009.37.68 during preparation. This view is an enlargement of the gastrolith grid seen in (A). Source credit: Sue Sabrowski, Royal Tyrrell Museum of Palaeontology.

*Wapuskanectes* specimen TMP2012.50.1 (Fig. 6) also comes from the Wabiskaw Member of the Clearwater Formation, but was found during road construction for the Parson's Creek Interchange on the north side of the town of Fort McMurray in December of 2012. It was located about 30 km south-southeast of the localities of the earlier specimens from the Syncrude mine. Unfortunately, it was exposed when a grader ploughed through the neck resulting in the destruction and loss of the skull and most of the neck. Additionally, there was no trace of the right forelimb, and this loss is interpreted to have occurred some time prior to the final burial of the animal. Despite these missing pieces, the

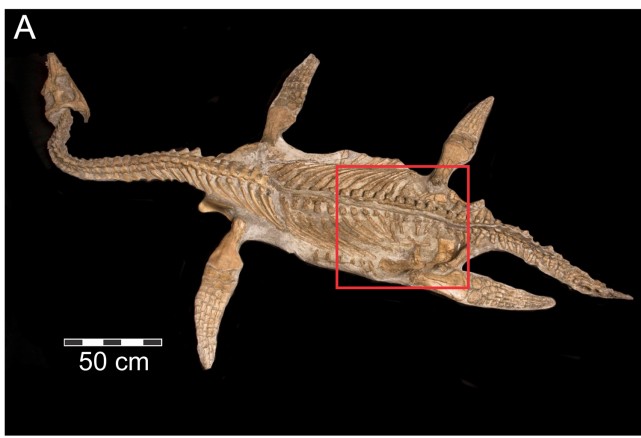

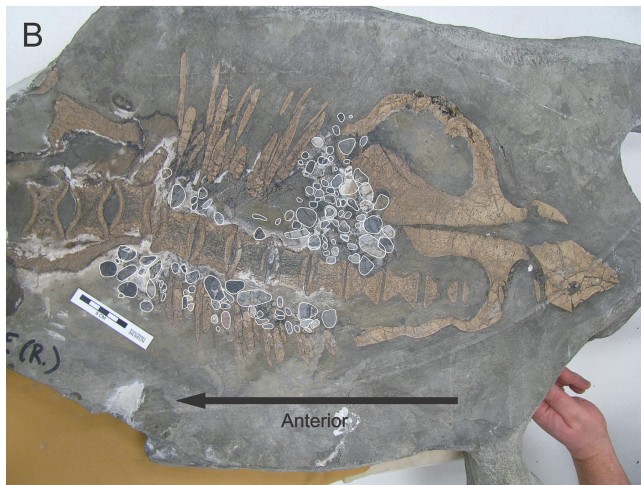

**Figure 4** *Nichollsaura borealis* **TMP1994.122.1** (*Druckenmiller & Russell, 2009*). (A) Dorsal view of the specimen which was found intact with its limbs fully outstretched. The left fore flipper, the phalanges of the right pes, the left proximal third of the cervical ribs, and the left anterior dorsal ribs are restored. The red rectangle shows the position of the gastrolith cluster visible on the underside. (B) Internal, ventral view of TMP1994.122.1 produced by a rock saw cut, originally intended to remove excess matrix, that fortuitously exposed the dorsal and ventral components of the axial skeleton and enclosed gastroliths (highlighted with white outlines). Source credit: Sue Sabrowski, Royal Tyrrell Museum of Palaeontology.

rest of the specimen is very well preserved. Preparation revealed some small gastroliths visible between the posterior dorsal ribs of this specimen. The relative completeness of this specimen enabled the generation of a 3D body model for mass estimation. The maximum transverse widths of the pectoral girdles of the two *Wapuskanectes* specimens, as measured at the postero-lateral margins of the glenoids, are identical at 57 cm, and it is assumed that the overall body sizes and proportions of the two animals would have been very similar.

## Stomach stones

*Albertonectes* was first exposed when an excavator bucket unexpectedly struck the fossil. Portions of the mid-trunk dorsal and ventral ribs, and an unknown quantity of gastroliths, were lost when the excavator bucket smashed through the fossil. Some time after death the carcass came to lie on the seabed on its back. As the stomach decayed the gastroliths

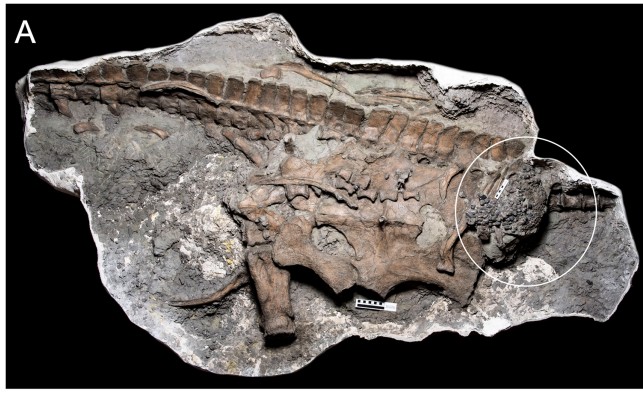

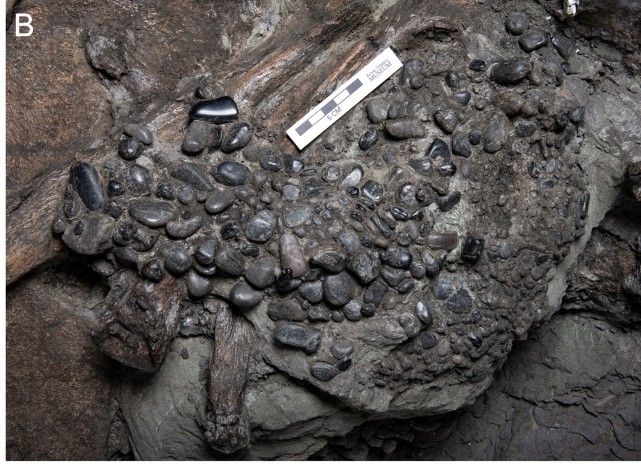

**Figure 5 An incomplete specimen of *Wapuskanectes betsynichollsae* TMP2011.88.1 that has a large number of gastroliths visible.** (A) Posterior cervical and anterior dorsal vertebrae in left lateral view, the pectoral girdle in ventral view and the left humerus, together with an intact collection of gastroliths located at the posterior end of the pectoral girdle (circled). Large scale bar at right glenoid is 10 cm. The maximum pectoral width of this specimen, 57 cm, is identical that of TMP2012.50.1. (B) Close up the gastroliths found with TMP2011.88.1. Source credit: Sue Sabrowski, Royal Tyrrell Museum of Palaeontology.              

gravitated downwards towards the medial side of the dorsal body wall and some came into direct contact with ribs. Subsequent compaction resulted in some portions of the ribs being plastically deformed around the gastroliths. Except for four stones that were knocked loose at the time of impact, the remainder were tightly clustered together and held in the matrix in a roughly spherical volume and did not show any degree of size or shape sorting. Given the uniqueness of the specimen, and its excellent state of preservation, it was decided to leave it intact and not remove any bones or gastroliths. A total of 97 gastroliths were revealed in total, four in the field, with the rest being exposed during preparation (Fig. 2B). A total of 52 stones could be reliably measured to obtain three dimensional radial measurements. The remaining 45 were visible but were either too deeply embedded in the hosting matrix, or were too obscured by ribs, to enable the collection of reliable geometry data. The gastroliths are all gray to black, sub-spherical, chert cobbles with some having a high polish.

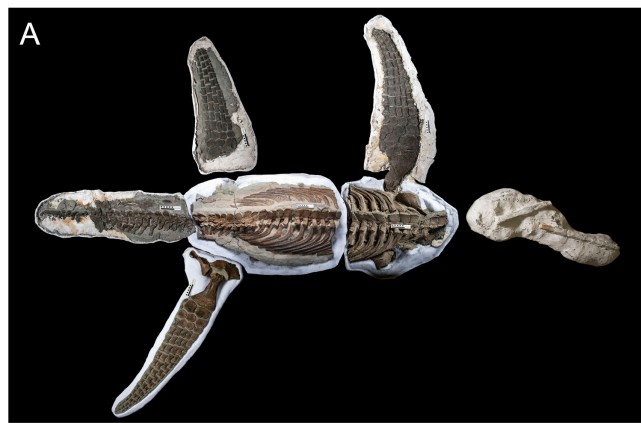

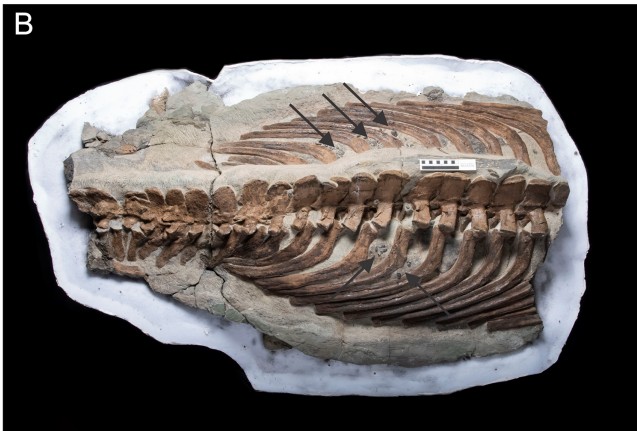

**Figure 6 A more complete specimen of *Wapuskanectes betsynichollsae*. TMP2012.50.1.** (A) Dorsal view of the originally upside down remains recovered from north of the town of Fort McMurray, Alberta. The head and most of the neck were lost when intercepted by a grader during road construction. The right flipper was lost sometime in the Early Cretaceous prior to the final burial of the body. (B) The pelvic and abdominal regions of TMP2012.50.1 showing gastroliths (black arrows) within the body cavity that settled between the ribs of the inverted carcass and hint at a more substantial collection that is hidden inside the body cavity. Scale bars on body regions are 10 cm. Source credit: Sue Sabrowski, Royal Tyrrell Museum of Palaeontology.

For *Fluvionectes*, given the incomplete nature of its preservation, and its modest level of disarticulation, it was decided to fully prepare its remains and remove all the bones. During this preparation 75 gastroliths was discovered and removed (Fig. 3B). These gastroliths tend to be disk-shaped and are all black cherts. The complete removal of these stones permitted the accurate measurement of their lengths, widths and depths, and the direct determination of their masses. These data were used as a basis for assessments of the reliability of the methods of estimation of missing gastrolith data used with the other plesiosaurs (see Methods).

The gastroliths of the *Nichollsaura* were inadvertently exposed when a saw cut, intended to cut away excess matrix from the ventral side of the block, went through the trunk region at a level that intersected limb girdles, dorsal vertebrae, gastralia and gastroliths (Fig. 4B). A total of 165 stones, composed of cherts and quartzites, are exposed on the cut surface as isolated, two-dimensional elliptic shapes with most of them arranged along the left and

right sides of the vertebral column, but with a substantial cluster immediately in front of the left pubis. This entire collection of stones appears to be spread out uniformly in a single layer. The visible co-occurrence of limb girdles, vertebrae, gastralia and gastroliths on the frontal plane defined by the cut surface demonstrates the substantial amount of dorso-ventral compaction that the fossil experienced after burial.

The gastroliths associated with *Wapuskanectes* specimen TMP2011.88.1 (Fig. 5B) were found during preparation as an isolated, elliptical cluster with semi-major and intermediate diameters of approximately 25 and 20 cm, respectively, and a thickness (semi-minor diameter) of roughly 5 cm. The stones are almost exclusively black, polished cherts with some minor quartzites. There are two regions within the cluster comprising two different sets of stone sizes: a loosely packed region of 93 stones with an average maximum diameter of about 2–3 cm and a densely packed region with an unknown, but large, number of small stones with an average maximum diameter of no more than 1 cm. This latter collection is defined by a sharp, posterior margin that may indicate the original boundary of the stomach (gizzard?) wall. The collection of coarser gastroliths was sufficiently exposed by mechanical preparation to obtain reliable lengths and widths for them, but not depths. It was not possible to obtain reliable measurements from any of the densely packed, smaller stones.

With the Fort McMurray Parson's Creek roadworks specimen of *Wapuskanectes*, TMP2012.50.1, there were no gastroliths visible initially. The specimen came to rest on the seabed on its back, and during initial preparation a single, black chert pebble was uncovered poking out between a pair of mid-trunk dorsal ribs on the right side of the body. Similar to what occurred with the *Albertonectes* gastroliths (see above), the stomach stones of TMP 2012.50.1 also settled towards the medial side of the dorsal ribs in the posterior trunk region. Before preparation of the torso went any further, the specimen was X-rayed at an industrial facility that analyses large metal castings for defects where they can easily detect tiny fractures and voids. The hope was that this technology would reveal the sizes and extents of other gastroliths within the body, but it was unable to detect anything resembling gastroliths (or even bones). It was conjectured that the enclosing matrix was too dense for the X-rays to penetrate sufficiently. Subsequent preparation revealed more small, black chert pebbles lying between the ribs, and close to the vertebral column, on both the left and right sides at a position just posterior to the midpoint of the trunk (Fig. 6B). Given the overall quality of the fossil, it was decided to leave all the bones in place, thus leaving an unknown number of gastroliths hidden within. Despite the limited number of gastroliths exposed with this specimen, they are very similar in size, shape and composition to the small gastroliths seen with TMP2011.88. As both specimens come from the same rock unit, the Wabiskaw Member of the Clearwater Formation, it was assumed that the overall gastrolith composition would be similar for the two specimens.

## METHODS

### Gastrolith volume and mass estimates

Four of the gastroliths associated with the *Albertonectes* were free of the matrix and could be confidently measured to obtain the three diameters, the remainder were partially

embedded within the matrix and some were also partly obscured by the ribs. Ninety seven stones were observed in total, but reliable measurements in all three dimensions could only be taken from 52 of them. These gastroliths were viewed as tri-axial ellipsoids and their semi-major, intermediate and semi-minor diameters were measured with digital calipers to the nearest tenth of a millimetre. The expression to compute the volume of a tri-axial ellipsoid is $Volume = \frac{4}{3}\pi abc$, where 'a', 'b', and 'c' are the semi-major, intermediate, and semi-minor radii, respectively, of the ellipsoid. Chert has a density of 2.67 g/cm$^3$ (*Deer, Howie & Zussman, 1992*), and the masses of these 52 chert gastroliths were computed by multiplying their volumes by the density of chert. For the remaining 45 stones only two diameters could be confidently measured. Lacking full knowledge of their dimensions, and noting that no sorting by shape or size was apparent with the fully measurable cluster of 52 stones, this latter assemblage was taken to be a representative, and random, sample of the original full set of gastroliths, and that remaining stone mass could be extrapolated from this sample. A simple ratio was used to estimate the combined mass of the remaining stones: if the measured 52 stones have a combined mass of 4.98 kg, then the mass of the remaining 45 will come from solving for '?kg' in the expression: $\frac{4.98\ kg}{52} = \frac{?kg}{45}$.

Only with *Fluvionectes*, and its fully removed gastroliths, was it possible to get the actual masses of all the stones. The stones were weighed to the nearest tenth of a gram on electronic scales (Symmetry ED-150 by Cole-Parmer). As a check on the gastrolith mass estimation schemes outlined above and below for the other plesiosaur specimens, the three, tri-axial dimensions of the Sage Creek gastroliths were also measured and their masses estimated by multiplying their calculated volumes by the density of chert. The semi-major and intermediate diameters were also used to predict the third diameter. The goal was to test the accuracy of the computed masses against the observed mass.

The *Nichollssaura* gastroliths were exposed on the saw-cut surface as closed contours of elliptic shape, from which the longitudinal and transverse diameters could have been obtained by direct measurement. However, to properly estimate the volume and shape of the gastroliths, the third diameter had to be estimated. It was decided to use a computational method to do this estimation and objectively determine the lengths and widths of the exposed elliptical cross-sections at the same time. A digital image of the cut surface exposing the gastroliths was imported into the drawing program Corel Draw! and all the visible perimeters of gastroliths were digitally traced, and the traces were then exported as individual AutoCAD DXF files. These DXF files were reduced to simple (X,Y) co-ordinate pairs defining the stone perimeters, and a computer program was written to carry out the following processing of the data. The longest diameter of a stone cross-section, 'a', was found using a computer program to determine the greatest distance between a pair of points for a given stone perimeter. Using its perimeter data, the cut-exposed area of the stone was computed using a triangular decomposition method (*Henderson, 2002*). The formula for the area of an ellipse is given by $Area = \pi\left(\frac{a}{2}\right)\left(\frac{b}{2}\right)$, where 'a' and 'b' are the semi-major and semi-minor diameters of the visible ellipse, respectively. Using the observed value for 'a' (the longest diameter) and the computed area, the value of

'b' can be derived *via* the expression $b = 2 \cdot \frac{Area}{\left(\frac{\pi a}{2}\right)}$. These values of 'a' and 'b' are used in estimation methods outlined in the next paragraph.

As seen with the *Nichollssaura* gastroliths, those of *Wapuskanectes* could only be reliably measured in two dimensions-length and width, with the third dimension, depth, needing to be estimated. Three methods were used to estimate this third dimension for the gastroliths of these two plesiosaurs: 1) depth = length ÷ 2; 2) depth = average of length and width = (length + width) ÷ 2; and 3) depth = square root of the product of length and width = √(length × width). These three methods were also applied to the fully measured stones of *Albertonectes* and *Fluvionectes* using the observed lengths and widths to estimate depths. These latter results were used to see which of the three methods produced an estimated mass closest to the properly measure ones.

For a long time mathematicians have been interested in knowing the best way to pack spheres so as to have the least amount of space between them, and this became known as the Kepler Conjecture (*Kepler, 1611*). Eventually, it was formally proven that the most efficient method achieves a packing density of approximately 0.74048 (*Hales, 2012*). The spherical packing principle was applied to the densely packed cluster of small gastroliths found with *Wapuskanectes* specimen TMP2011.88.1 in an attempt to estimate the mass of the stones. The gastroliths visible on the surface of the cluster are clearly touching all their neighbours, as posited by the sphere packing theorem, and this packing configuration was assumed to be true for the entire cluster of small stones inside. With an estimate of the volume of this cluster, multiplying the volume by the density of chert, and then applying the packing density factor would give an estimate of the cluster mass.

## Body mass estimations

Three-dimensional meshes representing the axial body and limbs of the four plesiosaurs were generated using the methods of *Henderson (1999)*. The actual body outlines were based on the preserved skeletal forms along with estimates of the soft tissues that would have surrounded them. For the neck regions, the dorso-ventral diameters were set so that space for an esophagus and trachea could be accommodated. Given the great length of the trachea in these animals, it was assumed that the tracheal lumen would have to be large to reduce the effects of the viscous drag on air flow caused by the large amount of surface area of the trachea. It has been shown that the resistance to flow decreases exponentially with increasing tracheal diameter (*Daniels & Pratt, 1992*). The outlines of the flippers were extended beyond their known bony extents by comparison with the limbs of sea-turtles and penguins. The bones forming the limbs of *Nicholssaura* and the *Wapuskanectes* road works specimen (TMP 2012.50.1) were both much broader (greater fore-aft chordal lengths) than those of the two elasmosaurs, and this resulted in their modelled limbs being broader than those of the elasmosaurs. Lung volumes were estimated for each model, and were set to represent between 8–10% of the axial body volume, based on observations of the lungs of living reptiles (*Tenney & Tenney, 1970*). Actual computation of the masses of the body regions (axial and appendicular) involves multiplying the volumes of these

regions by a predefined value for the tissue density. The volumes were estimated using the methods outlined in *Henderson (1999)*.

A uniform body density of 1,050 gm/L was used for all the post-cervical body regions in all the models. This value is higher than that of pure water at 1,000 gm/l, but it was felt that given the dense, non-pneumatic nature of the bones of the axial and appendicular skeletons, and in particular the high proportion of bone relative to soft-tissue inferred for the limbs, that this denser value was justified. The necks, with their large, air-filled trachea, were set to a lower density of 900 gm/l. Using *Albertonectes* as an example, this latter value comes from a simple analysis of the cross-sectional area of the neck of the *Albertonectes* model as measured at the mid-neck position and the cross-sectional area of an air-filled trachea with a diameter of 15 cm. The depth of the neck at midpoint is 50 cm, so the tracheal diameter is just 30% of that. The cross-sectional area of the mid-neck is 0.173 m$^2$, while that of the trachea is 0.0177 m$^2$. The presence of a trachea with this diameter would reduce the cross-sectional area of the flesh-and-bone-filled neck by approximately 10%. Extending this areal reduction along the length of the neck results in a mass reduction of 10% for the neck and reduction in its density by 10% from the basic tissue value of 1,050 gm/l to approximately 900 gm/l. Despite the main body density value being greater than that of water, when a full body model is combined with its lung, the animals are still positively buoyant when the lungs are fully inflated as observed in a study of buoyancy and equilibrium in plesiosaurs that used digital models with the same densities and lung volume proportions used here (*Henderson, 2006*). The assumptions and methods that went into the generation of the above models have been validated with accurate mass estimates for similarly constructed models of extant, semi-aquatic reptiles—alligators (*Henderson, 2003*), sea turtles (*Henderson, 2006*) and penguins (*Henderson, 2018*)–and fully terrestrial animals such as horses and giraffes (*Henderson & Naish, 2010*) and elephants (*Henderson, 2004*).

## RESULTS

Figure 7 shows the four body models generated from the plesiosaur skeletons, and Table 1 presents the physical attributes determined from the models. Two of the models were based on nearly complete fossils–*Albertonectes* and *Nichollssaura*–and their necks (grey coloured regions on models) are estimated to weigh 1,100 and 10.6 kg, respectively. Relative to body mass, these necks represent 22.7% and 7.86% of total body mass. The necks of the other two–*Fluvionectes* and *Wapuskanectes*–were incompletely preserved, and this leads to a small uncertainty regarding their actual body mass. As restored, the necks for these two animals are estimated to have mass of 72.9 and 198 kg, respectively, and their relative mass fractions are 18.6% and 16.5%, respectively. As these latter two animals are early elasmosaurids from close to the Aptian/Albian boundary, they were given substantial necks and their relative neck mass fractions approach that of the later and derived Campanian *Albertonectes*.

Table 2 presents observed and estimated gastrolith masses for the four plesiosaurs. Except for *Fluvionectes* where the actual amount of stomach stones is known exactly, these estimated values will have to be considered as minimums. This is an unavoidable fact

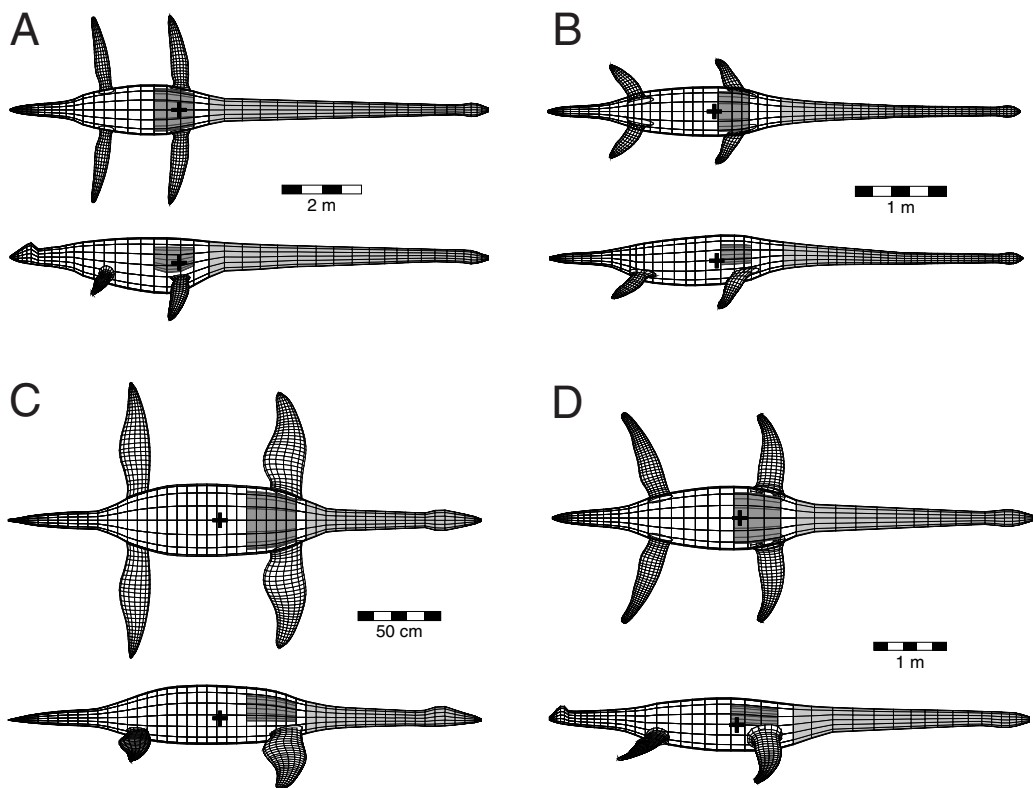

**Figure 7 Three-dimensional, digital models of the four plesiosaurs analyzed in the present study.** The models were used for calculation of body mass estimates and were generated by the author based on examination of the fossils held at Royal Tyrrell Museum of Palaeontology. (A) *Albertonectes vanderveldei* TMP2007.11.1. 9 (B) *Fluvionectes sloanae* TMP2009.37.68 with body shape based on the outlines presented in *Campbell et al. (2021)*. (C) *Nichollssaura borealis* TMP1994.122.1. (D) *Wapuskanectes betsynichollsae* TMP2012.55.1. Hollow lung volumes are indicated by the dark grey shapes in the chest regions. Limb and axial body densities were set to 1,050 gm/l, except for the necks (medium grey) which were set to 900 gm/l to account for the empty space occupied by the tracheal and esophageal volumes. Black '+' signs indicate the centres of mass computed for the combined axial body, lungs and limbs and incorporates the regional density differences. See Table 1 for the physical parameters determined for the models.

**Table 1 Summary of physical attributes of the four plesiosaur models.** Ordering is alphabetical by genus.

| | Body length (m) | Total body mass (kg) | Lung volume (L)[1] | Axial body mass (kg) | Single arm mass (kg)[2] | Single leg mass (kg)[2] | Centre of mass (m)[3] |
|---|---|---|---|---|---|---|---|
| *Albertonectes vanderveldei* TMP2007.11.1 | 12.1 | 4,810 | 43.8 (9.36%) | 4,480 | 115 (2.40%) | 52.0 (1.08%) | (4.26, 0.579) |
| *Fluvionectes sloanae* TMP2009.37.68 | 5.21 | 392 | 24.2 (6.20%) | 378 | 3.42 (0.875%) | 3.54 (0.905%) | (1.84, 0.274) |
| *Nichollssaura borealis* TMP1994.122.1 | 2.86 | 135 | 10.7 (8.27%) | 117 | 2.75 (2.17%) | 2.19 (1.72%) | (1.26, 0.163) |
| cf. *Wapuskanectes* sp. Plesiosaur TMP2012.50.1 | 6.63 | 1,120 | 84.3 (7.31%) | 1,090 | 32.0 (2.68%) | 19.8 (1.67%) | (2.57, 0.356) |

**Notes:**
[1] Lung volume percentage expressed as a percentage of total axial body volume.
[2] Limb mass percentage expressed as a percentage of total body mass.
[3] Centre of mass expressed as (X,Y) where X is the distance from tip of tail and Y is the height above the ventral-most part of the axial body.

**Table 2  Observed and estimated gastrolith data for the plesiosaurs in the present study.** See methods and results for derivations of predicted missing mass. Ordering is alphabetical by genus. Total gastrolith mass as a percentage of total body mass (Table 1) is shown in parentheses.

| | Measured/observed mass (gm) | Predicted missing mass (gm) | Total gastrolith mass (gm) | Largest single gastrolith (gm) |
|---|---|---|---|---|
| *Albertonectes vanderveldei* TMP2007.11.1 | 4,980 | 4,310 | 9,290 (0.193%) | 1,130 |
| *Fluvionectes sloanae* TMP2009.37.68 | 359 | † | 359 (0.0916%) | 15.3 |
| *Nichollssaura borealis* TMP1994.122.1 | †† | 105 | 105 (0.0778%) | 9.96 |
| *Wapuskanectes* sp. TMP2012.50.1 | †† | 497 | 497 (0.0444%) | 14.5 |

Notes:
† All *Fluvionectes* stones could be weighed, so no estimate needed.
†† None of the stones were fully visible, so no sets of observable triaxial measurements possible.

resulting from two factors: the lack of knowledge of the true number of stones hidden in the matrix in the specimens, and the unknown number of stones that may have been lost at the time of discovery. However, it is reasonable to assume that the number of stones within the bodies of living plesiosaurs will be highly variable between species and even from year to year in the same individual. The 52 stones that could be fully measured for *Albertonectes* were found to have a combined volume of 1,865 cm$^3$, and when multiplied by the density of chert at 2.67 g/cm$^3$, to have a mass of 4,980 g. From the calculated mass of the observable 52 stones, the remaining, unmeasured 45 stones were predicted *via* a simple ratio expression to have a combined mass of 4,310 g. Together, these two estimates gave a total gastrolith mass of 9,290 g. Using the measured semi-major and intermediate diameters of the first 52 gastroliths of *Albertonectes*, the three methods of predicting the third diameter gave the following mass results-Half Length: 4,654 gm; Average of Length and Width: 10,695 g; square root of the product of Length and Width: 10,577 g. The Half Length method clearly produced the result closest to the one that used the three measured diameters.

Direct weighing of the *Fluvionectes* gastroliths gave a total gastrolith mass of 359 gm. Determining the volumes of the stones from their measured diameters, and multiplying them by the density of chert, gave a total mass estimate of 325 gm–a value 9.56% less than the directly weighed value. At first it might seem reasonable to use the 9.56% mismatch between the observed and predicted stone masses for *Fluvionectes* as a correction factor to be applied to the stone masses calculated for *Albertonectes* that were also based on geometry. However, the shape distributions of the two assemblages are very different (see the histograms and pie charts below), suggesting that the mismatch between observed and calculated mass for *Fluvionectes* might be sample specific, and could not be applied as a correction factor to the results obtained for *Albertonectes*. Applying the three, third-dimension estimation methods to the measured semi-major and intermediate diameters of the *Fluvionectes* gastroliths gave the following mass estimates-Half Length: 337.7 gm; Average of Length and Width: 780.7 gm; square root of the product of Length and Width: 770.8 gm. Again, the Half Length estimation provides the estimate closest to the one derived from actually weighing the gastroliths.

In light of the close agreement between gastrolith mass estimates produced using the Half Length method and those from direct measurement of *Albertonectes* and *Fluvionectes*

gastroliths, the obscured depth dimensions for the gastroliths of *Nichollssaura* and *Wapuskanectes* were estimated using the Half Length method. This method gave a total mass for the cut *Nichollssaura* gastroliths of 105.4 gm, and a mass of 186.7 gm for the scattered cluster of larger gastroliths from *Wapuskanectes*.

The densely packed cluster of small gastroliths found with *Wapuskanectes* specimen TMP2011.88.1 has a shape similar to that of a flattened, tri-axial ellipsoid with its length, width and depth being 20, 10 and 5 cm, respectively. This cluster has a volume of 157.1 cm$^3$, and when multiplied by the density of chert and the spherical packing density factor, has an estimated stone mass of 310.6 gm. Adding the mass estimates of the loose and dense clusters give a total mass of 497.3 gm.

Histograms and pie charts were generated to aid visualization of the differences and similarities between the shapes and masses of stomach stones. Figure 8 presents histograms that show marked differences in the ranges and proportions of the masses of stones ingested by the animals. *Albertonectes*, being the largest animal in the study, also had the greatest range of stone sizes, with the largest stone being 1.13 kg. The largest stones recovered from the other animals were all an order of magnitude smaller (Table 2). To facilitate comparisons between the stones of all four animals, the histogram for *Albertonectes* was done again but with the same smaller range of bin sizes (0–16 gm) that was applied to the other specimens (Fig. 8B). The gastroliths of the three smaller animals are dominated by stones in the 0–4 gm range, while these stones represent a minor component for *Albertonectes*. Visual inspection of the gastrolith size distributions for the smaller animals shows them to be skewed to the right, but with *Fluvionectes* having less of a skew.

Pie charts were generated for all the gastrolith samples using four shape categories–xiphoid ("blade-like"), discoid, spherical, and cylindrical based on the classification scheme of *Dobkins & Folk (1970)*. The pie charts only analyze those gastroliths that could be fully or partly measured. The pie chart proportions were computed two ways: 1) using the individual masses of the stones and seeing how the masses of stones distributed across each category (Fig. 9), and 2) the absolute count of how many stones were in each category (Fig. 10). For *Albertonectes* and *Fluvionectes*, their pie charts demonstrate that the two different ways of looking at gastrolith shape distribution can give very different results. This is especially noticeable with the blade-like (xiphoid) class where *Albertonectes* would appear to have almost no stones in this category (1.1%) when their mass is considered (Fig. 9A), but almost 10 times as much when counted (Fig. 10A). However, both measures are consistent in showing that the blade-like gastroliths of *Albertonectes* make up the smallest fraction of the entire collection. The differences in shape category fractions based on different methods of analysis also apply to *Fluvionectes* (Fig. 9B *vs*. Fig. 10B). It should be noted that the present count fraction pie chart for *Albertonectes* gastroliths differs slightly from the one originally presented in the initial description by *Kubo, Mitchell & Henderson (2012)*, Fig. 9. This is the result of a small error the earlier version of the computer program used for calculating the shape fractions. The corrected plot now has slightly larger spheroid and smaller cylindrical fractions than were previously presented.

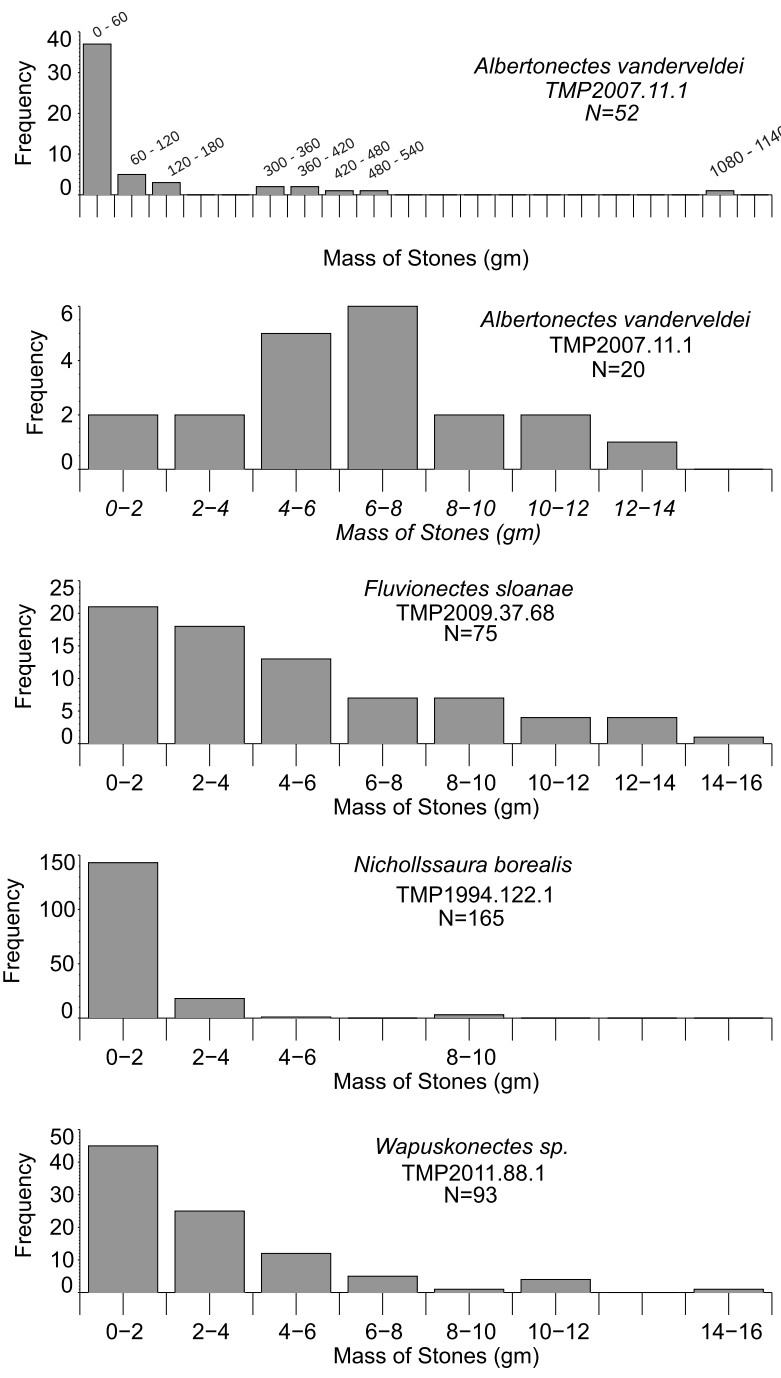

**Figure 8 Histograms showing the very different mass distributions of the sets of gastroliths found with the plesiosaurs of the present study.** Note that the *Albertonectes* analysis is done twice: first encompassing the full mass range of its stones, and second only applying the smaller mass range of 0–16 gm that was also used with the other plesiosaurs. Only the stones masses of *Fluvionectes* were weighed directly. The other masses came from estimation techniques. See Methods and Results for details. Note that the maximum frequency range of each histogram plot is different to make the detailed counts visible.                                     

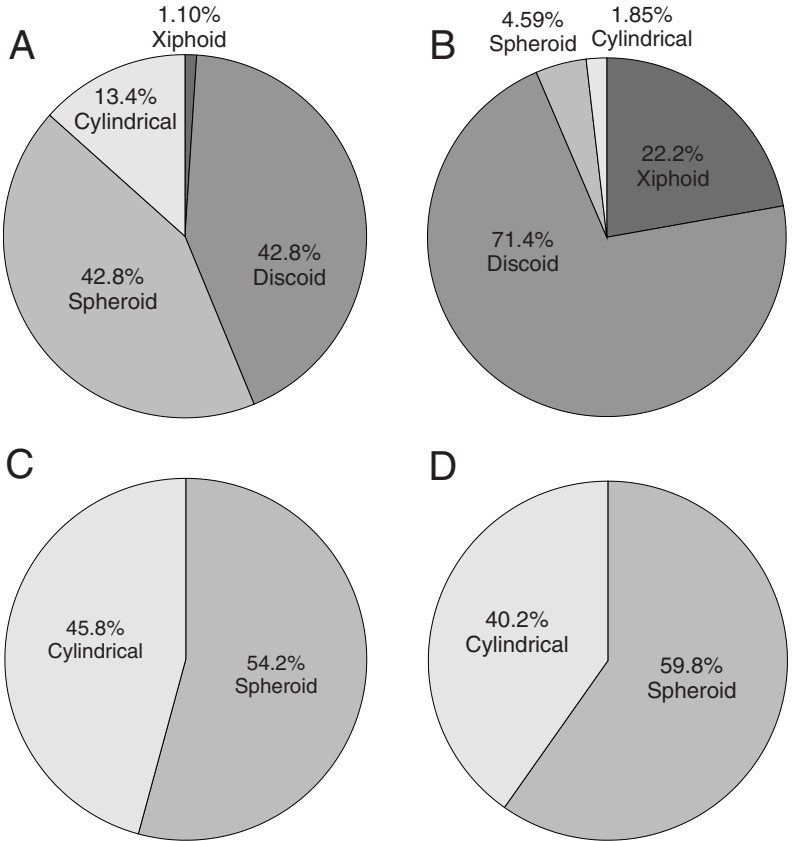

**Figure 9** **Pie charts showing the relative proportions by mass of the different shapes of the gastroliths recovered from the four plesiosaur specimens using the shape categories of *Dobkins & Folk (1970)*.** (A) *Albertonectes vanderveldei* TMP 2007.11.1, Bearpaw Fm., Upper Cretaceous, southern Alberta. Total mass = 4,980 g. (B) *Fluvionectes sloanae* TMP 2009.37.68, Dinosaur Park Fm., U. Cretaceous, southern Alberta. Total mass = 359 g. *Fluvionectes* is distinguished from the others by the high fraction of discoid gastroliths and may reflect its being found in fluvial sediments. (C) *Nichollssaura borealis* TMP 1994.122.1, Clearwater Fm., L. Cretaceous, northern Alberta. Total mass = 105.3 g. (D) cf. *Wapuskanectes* sp.TMP 2011.88.1, Clearwater Fm., L. Cretaceous, northern Alberta. Total mass = 186.7 g. *Nichollssaura* and *Wapuskanectes* both come from the same formation and show very similar gastrolith shape distributions that are dominated by spherical pebbles.     

Figure 11 summarizes the relationships between largest gastrolith and total gastrolith mass relative to body mass. The two orders of magnitude differences seen in the ranges of stone masses necessitated plotting the data on log-log plots. Although it is just a sample of four plesiosaurs, and the gastrolith collections each have their deficiencies, two trends are noticeable from the two plots. The first is that the largest plesiosaur, *Albertonectes*, has both the largest single gastrolith and the most gastroliths. A regression computed for the largest gastrolith *vs.* body mass has a least squares correlation coefficient of 0.8762, but the uncertainty on the computed slope (0.5474) is almost half the slope value (1.4076). A better regression result comes from plotting total gastrolith mass against body mass.

The uncertainty on the computed slope (0.2446) is less than half that seen with previous plot, and the correlation coefficient is much higher at 0.9606. In both plots, the amounts of gastroliths estimated for *Wapuskanectes* lie below the fitted line. The collection of stones

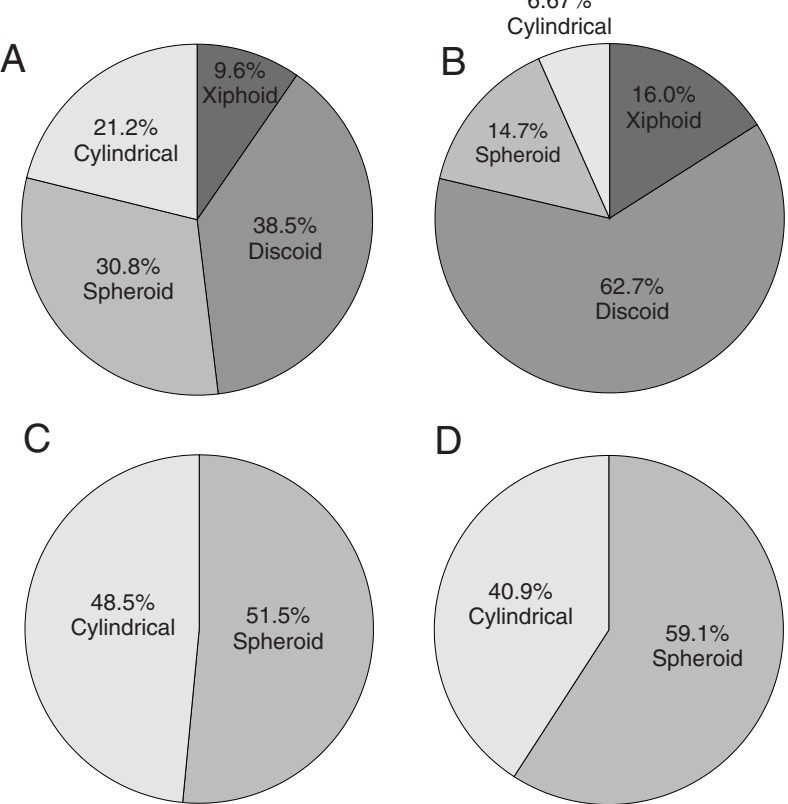

**Figure 10** Pie charts showing the relative proportions by absolute counts of the different shapes of the gastroliths recovered from the four plesiosaur specimens using the shape categories of *Dobkins & Folk (1970)*. (A) *Albertonectes vanderveldei* TMP 2007.11.1, Bearpaw Fm., U. Cretaceous, southern Alberta. Number of stones = 52. The higher number of xiphoid clasts with *Albertonectes* contrasts with their lower total mass fraction seen in Fig. 9, and shows that these clasts tend to be small. (B) *Fluvionectes sloanae* TMP 2009.37.68, Dinosaur Park Fm., U. Cretaceous, southern Alberta. Number of stones = 75. *Fluvionectes* still shows a majority of discoid gastroliths. (C) *Nichollssaura borealis* TMP 1994.122.1, Clearwater Fm., L. Cretaceous, northern Alberta. Number of stones = 165. (D) cf. *Wapuskanectes* sp. TMP 2011.88.1, Clearwater Fm., L. Cretaceous, northern Alberta. Number of stones = 93. *Nichollssaura* and *Wapuskanectes* continue to show the same pattern of having mainly spherical gastroliths.

found with *Wapuskanectes* TMP2011.88.1 was disturbed after death as shown by the "smeared-out" shape of the collection, and suggests that an unknown number of gastroliths were lost prior to final burial of the carcass. Of the four fossils studied, this one was also the most disrupted, with the pectoral girdle being inverted relative to the rest of the body prior to final burial. The specimen was discovered when some neck vertebrae were exposed by heavy equipment. It was removed from the mine as a single, large, intact block with no chance of any stones being lost during this process (D Henderson, 2011, personal observation).

## DISCUSSION

The two Late Cretaceous elasmosaurs examined in the present study, *Albertonectes* and *Fluvionectes*, come from rock formations that represent very different environments-open

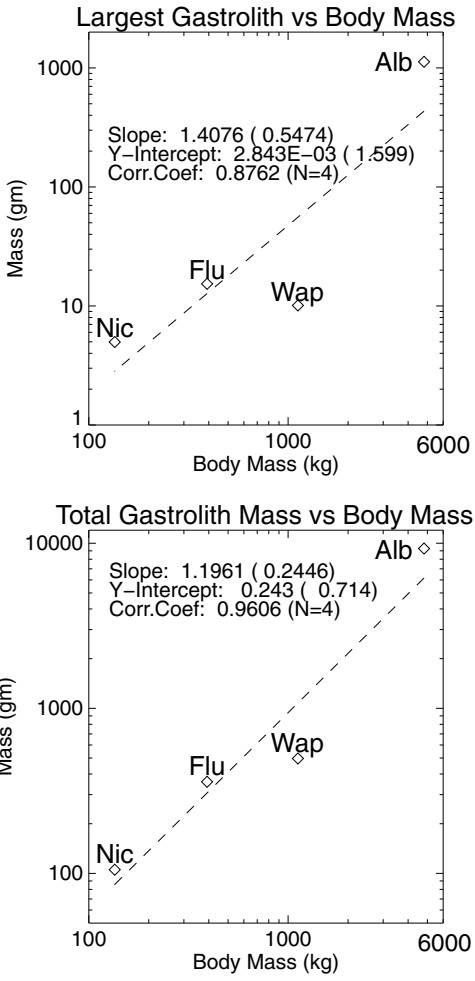

**Figure 11 Maximum plesiosaur gastrolith size and total gastrolith mass plotted as functions of body mass for the four animals of the present study.** Although based on a sample of just four animals, larger plesiosaurs appear to have a larger maximum stone size and greater total mass of stones than smaller plesiosaurs. See Table 1 for body mass data and Table 2 for stone mass data. Logarithmic plotting is used as both body masses and stone masses span approximately two orders of magnitude. Abbreviations: Alb, *Albertonectes vanderveldei;* Flu, *Fluvionectes sloanae;* Nic, *Nichollssaura borealis;* Wap, *Wapuskanectes* betsynichollsae.                     

marine *vs.* fluvial, respectively, and this could explain the differences in their respective gastrolith compositions. This idea is supported by the observation that *Nichollssaura* and *Wapuskanectes*, both from the Wabiskaw Member of the Clearwater Formation, have very similar gastrolith compositions. This is also remarkable because the animals are very different in body proportions and size, with *Wapuskanectes* being approximately ten times more massive than *Nichollssaura* (Table 1).

   The pie charts reveal that the discoid fractions of the stomach stones represent either major (*Fluvionectes*) or substantial (*Albertonectes*) portions of the sample. The strong discoid signal seen in these samples suggests that these animals were collecting potential stomach stones mainly from beaches (discs), but also some from the mouths of rivers (spheres), based on the observations of *Dobkins & Folk (1970)*. The bias towards crypto-

and micro-crystalline quartz (chert) gastroliths for plesiosaurs of the present study may indicate that these animals actively sought these rock types for their hardness and durability. It could also be argued that we have biased samples of stomach stones because any ingested limestone or carbonate-cemented clastic pebbles would have been dissolved and/or disaggregated by stomach acids and the small constituent grains excreted. However, if the gastrolith samples represent an instantaneous, random "snap-shot" of the stone assemblages carried by the animals at the time of their deaths, it is surprising that none of four gastrolith samples studied show even the slightest trace of any sandstone or limestone pebbles.

Table 2 shows that the masses of gastroliths represent extremely small fractions of total body mass, with an average value being just one tenth of one percent. The 9.29 kg mass of stones estimated for *Albertonectes* is similar to the 8.84 kg mass of stones recovered from an elasmosaur from the Bearpaw Formation of Montana (*Darby & Ojakangas, 1980*), but the largest stone from this latter animal was just 662 gm, approximately half the size of the largest stone from *Albertonectes*. In their review of elamosaur gastroliths, *Thompson & Martin (2007)* comment on the extremely variable nature of the range of sizes and the total masses of the gastroliths recovered from this group of plesiosaurs. A large Antarctic specimen examined by *Thompson & Martin (2007)* was found to have 2,626 gastroliths, but their total mass was just 3.02 kg, and the largest was only 46.6 gm. At the other extreme, *Everhart (2000)* reports on a total mass of gastroliths from a Kansas elasmosaur being equal to 13.1 kg with the largest weighing 1.49 kg. If gastroliths were truly related to buoyancy control the expectation would be that elasmosaurs of the same size should have the same amounts of gastroliths, but this is clearly not the case. Similar to what was argued for in modelling studies of gastroliths in crocodilians (*Henderson, 2003*) and plesiosaurs (*Henderson, 2006*), and the suspicions of other authors (*e.g.*, *Cicimurri & Everhart, 2001*), the effects of these relatively tiny, and variable, masses of stones on the buoyancy and equilibrium of the animals when they were alive are considered to have been almost non-existent.

The stones found with *Nichollssaura* and *Wapuskanectes* are very similar in size, shape and composition to the clasts found within the Cadomin Conglomerate, a Lower Cretaceous rock formation which is found throughout the length of the Rocky Mountain foothills in Alberta from the 49th to the 60th parallels (*Schultheis & Mountjoy, 1978*). This identification was done with reference to a large sample of the conglomerate held in the collections of the Royal Tyrrell Museum with specimen number TMP2016.38.4. This conglomerate is early Aptian in age, while the Wabiskaw Member and its plesiosaurs are earliest Albian, and this temporal gap represents an interval of approximately 8–10 million years (*Alberta Geological Survey, 2015*). It is hypothesized that *Nichollssaura* and *Wapuskanectes*, living the northern Western Interior Seaway during the Albian, could have swum westwards to beaches and river mouths along the west coast of the Seaway to collect either uncemented, or eroding, Cadomin clasts.

## CONCLUSIONS

Using samples of gastroliths from four plesiosaur fossils (two Early Cretaceous and two Late Cretaceous) with varying degrees of completeness, it has been shown that plausible minimum estimates of the original masses of the stones can be made. Total body masses were made for the four animals, and these can be used as a reference standard for other plesiosaurs with similar body shapes. Consistent with earlier suspicions about the ineffectiveness of the relatively small amounts of stones found with plesiosaurs on their buoyancy, the total masses of the stones found with the present plesiosaurs was never more 0.2% of total body mass. With the geometry data measured from the stones their shapes can be categorized and hypotheses made about where the animals collected the stones, *e.g.* river or beach setting. Lastly, the sizes, shapes and compositions of the gastroliths found with the two different species of Early Cretaceous plesiosaur that came from the same geographic area and time were found to be very similar, suggesting that the animals collected their stone assemblages from the same area.

## INSTITUTIONAL ABBREVIATION

TMP        Royal Tyrrell Museum of Palaeontology.

## ACKNOWLEDGEMENTS

The assistance from the Syncrude Canada Ltd. mine people when we regularly arrive to collect the fossils they bump into (*Nichollssaura* and *Wapuskanectes*, in this case) is greatly appreciated, as their people always go out of their way to help us. Similarly, the staff at the Korite Mine were most helpful when we collected *Albertonectes*. Many people who were, or still are, staff at the Royal Tyrrell Museum of Palaeontology (RTMP) were involved in the lengthy preparations of the fossils discussed in this study. Preparation of *Albertonectes* was done by Mark Mitchell, Robin Cooke, Dawna Macleod and Judy Graham. *Fluvionectes* was prepared by Rebecca Sanchez. *Nichollssaura* was prepared by Mark Mitchell, Shane Zimmer, Stuart Wright and Jim McCabe, with restoration work done by Mark Mitchell. *Wapuskanectes*specimen TMP2011.88.1 was prepared solely by Mark Mitchell, while the Parson's Creek specimen,. TMP2012.50.1, was prepared by Marilyn LaFramboise, Ramon Nagesan, Emily Frampton and Samantha Haddon. The X-raying of the torso block of the the Parson's Creek *Wapuskanectes* specimen was done in Burnaby, British Columbia at the invitation of David G. Stasuk, president of Stasuk Testing & Inspection Ltd. Mr Stasuk and his staff put in many long hours and a whole weekend in an attempt to get results. Rebecca Sanchez and Tom Courtenay of Collections at RTMP dealt with my many requests to move and view specimens. Specimen photography and assembly of composite images was done by Sue Sabrowski (RTMP). Mark Mitchell (RTMP) is thanked for his expertise in the field when we were collecting three of the specimens and for his profound knowledge of the plesiosaurs in the RTMP collections.

### Funding

Article processing charge paid for by the Royal Tyrrell Museum Cooperating Society. The funders had no role in study design, data collection and analysis, decision to publish, or preparation of the manuscript.

### Grant Disclosures

The following grant information was disclosed by the authors:
Royal Tyrrell Museum Cooperating Society.

### Competing Interests

The author declares that he has no competing interests.

### Author Contributions

- Donald M. Henderson conceived and designed the experiments, performed the experiments, analyzed the data, prepared figures and/or tables, authored or reviewed drafts of the article, and approved the final draft.

### Data Availability

The data and code are available in the Supplemental Files.

### Supplemental Information

Supplemental information for this article can be found online at http://dx.doi.org/10.7717/peerj.17925#supplemental-information.

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
