# Peer review of "Lost, hidden, broken, cut-estimating and interpreting the shapes and masses of damaged assemblages of plesiosaur gastroliths"

_PeerJ, doi:10.7717/peerj.17925_

## Round 0.1 · original submission · Major Revisions

Dear authors,

The three reviewers have not come to a common recommendation, so I have accepted the majority opinion of 'major revisions'.

Please note that the reviewers have commented on the grammar and phrasing of your submission. PeerJ does not provide linguistic services as a standard service, therefore any changes will have to be made prior to resubmission.

Overall all three reviewers were very complimentary about your manuscript. All three reviewers highlighted slightly different concerns, or potential other aspects to consider. Reviewer three in particular.

I look forward to receiving your revised manuscript.

**Language Note:** The Academic Editor has identified that the English language must be improved. PeerJ can provide language editing services - please contact us at [email protected] for pricing (be sure to provide your manuscript number and title). Alternatively, you should make your own arrangements to improve the language quality and provide details in your response letter. – PeerJ Staff

·

Basic reporting

This paper aims to describe the sets of gastroliths in three elasmosaurid species and a leptocleidid species from Alberta, Canada. Most of the description ultimately serves to reject previous hypothesis suggested for the function of the gastroliths (e.g., ballast hypothesis); but no alternative hypothesis is presented. The shapes, location, masses and counts of the sets of gastroliths are presented and discussed relative to each other. The scope of the paper as written is solely focused on the specimens analysed, but no effort to examine the plesiosaur gastrolith record was made. Perhaps that is my main criticism: the new data could be contextualized in a broader phylogenetic framework, such that the paper could be useful to infer broader patterns among plesiosaurs. The available information in the literature could allow for inferring these broader patterns. In one (two) instances, a probable source of the stones was identified.

Literature on the interpreted paleonenvironments is lacking.

The results circumspect the hypothsis, hence inferences are inherently local, yet, valid and interesting in their own right.

Experimental design

The methods are described in sufficient detail, however, in some sections some assumptions are questionable. The research is original and fits the scope of the journal. The research question is well defined: what can we say about the gastroliths in four species of Canadian Cretaceous plesiosaurs?

Validity of the findings

This study is interesting and rare, hence valid and regionally/phylogetically valid. The author tried to understand as much as possible the omissions and complexities of concealment and loss of gastroliths in the analysed fossils.

Reviewer 2 ·

Basic reporting

The scientific content of this paper is clear enough to follow. I have no issues with the scientific content.

However, some sections of the manuscript do not quite meet the standards of good quality, professional English writing. Some of the phrasing is imprecise, poorly constructed and on occasion grammatically incorrect.

In the uploaded edited file, I have indicated some of the places where I think improvements should be made. However, I would strongly suggest running the entire text through a grammar/language checker or using an editing service to improve the style.

Experimental design

Research question is clear and results are well presented. There are many aspects of this work that are good, precise and interesting.

Validity of the findings

In general, I think this paper makes a good contribution to gastrolith research but I think the overarching meaning of the results relative to other work could be given more emphasis.

These results add to previous work, providing evidence against certain functional speculations, for example, the buoyancy theory and I think this is important and could be stated more strongly.

Also no correlations are made between gastrolith content and dietary preferences in the four species of plesiosaurians. Is that something that could be added/commented on?

Additional comments

The scientific content, data analysis and figures are well set out and easy to follow. Diagrams and figures are clear. Some suggested corrections to captions are indicated in the uploaded edited paper.

Importantly, the paper shows how plausible estimates of gastrolith content can be made, even in specimens where the gastrolith content is only partially visible/measurable. The detailed and precise methodological report contributes usefully to gastrolith studies.

To my mind the most general and important conclusion of this paper is that contra Taylor (1993), gastrolith mass and composition do not correlate precisely with plesiosaurian body mass or shape. Thus, gastrolith content is unlikely to be related to buoyancy control. It is surprising that this is not stated as a concluding statement in the Abstract and I think it is important to include it.

The author argues that if gastroliths were involved in buoyancy regulation, elasmosaurids of the same size would be expected to harbour a similar gastrolith mass (line 471) which they do not. Similarly, plesiosaurians of very different sizes and shapes (like Nichollssaurua and Wapuskanectes) would not be expected to have similar gastrolith compositions or masses (lines 437-442), which they do (Table 1). The paper also supports the conclusions of other authors who argue that the effects of the calculated gastrolith content relative to the plesiosaurian body masses would be minimal. The gastrolith masses in the four specimens studied never exceeded 0.2% of the body mass, but were otherwise quite variable, ranging from 0.04 % of the total body mass in Wapuskanectes to 0.19% in Albertonectes.
I think these are interesting conclusions, supported by good data.

Annotated reviews are not available for download in order to protect the identity of reviewers who chose to remain anonymous.

Reviewer 3 ·

Basic reporting

Writing is superb with ideas and methods well-explained. Nicely made figures. Some grammatical errors are present and rephrasing of some sentences are needed however.

Experimental design

The experimental design is overall good but there were some assumptions made that need to be addressed. Principally, there were three 'levels' of assumptions: 1) The count for the number of gastroliths in each plesiosaur specimen were based on gastroliths that are visible. However, three of the four studied plesiosaurs are not completely prepared. Thus, I was concerned that the actual number of gastroliths per plesiosaur might in fact be greater than what is observed because the remainder are obscured by matrix or overlying bones. It was mentioned in the text that some gastroliths were lost during initial discovery and this skews the data of course with a lower number of gastroliths being reported than were initially present. This is of course a major issue if the idea is to determine the total number of gastroliths present and their mass relative to the plesiosaurs. 2) To estimate the mass of the gastroliths, the density of chert was used in the calculation, however, not all gastroliths were composed of chert (as in the case of Nichollssaura and one of the Wapuskanectes specimens). The heterogeneity of the gastroliths I feel needs to be taken into account when calculating the gastrolith masses. 3) It was really nice to see the models of the plesiosaurs made with an attempt to calculate their respective masses. This is laudable as there is next to nothing today in the literature on body masses for plesiosaurs :) However, it would be good to see more information in the methodology as to how these models were constructed. For example, Albertonectes is booked at 11.2 meters (Kubo et al., 2012) but reconstructed as 12.1 meters by Henderson. Where did the extra 0.9 meters come from? Is this extra length from adding a skull? I also see for Albertonectes that the trunk is quite deep relative to mediolateral width, thus making a deep ellipsoidal shape. Is this an accurate reconstructions of an elasmosaurid torso? There is a paper out by O'Keefe et al. (2011) on Tatanectes and it briefly reviews body shapes of plesiosaurs. This might be a good paper to reference for body reconstructions. Welles (1943) also provides some torso reconstructions. Although at the end of the day, nobody really knows what the body shape of an elasmosaurid is... some motivation for the reconstructions and how they were devised needs expansion.

Validity of the findings

The validity of the findings is contingent on the assumptions made for the number of gastroliths present in each plesiosaur, the mass of the gastroliths, and the mass of the plesiosaurs based on the models which needs to be addressed.

Lastly, in the discussion it is explained that it is unlikely the gastroliths were used as a ballast but no alternative explanation is provided leaving the reader with something to be desired.

Additional comments

Thank you very much for an extremely well-written manuscript. If you motivate your reasoning for the assumptions made in the methodology and go deeper into why you designed the plesiosaur models the way you did (which of course has implications for the mass estimates) this will surely improve the manuscript. Find enclosed an annotated PDF of the manuscript with some comments and improvements to writing and grammar.

Annotated reviews are not available for download in order to protect the identity of reviewers who chose to remain anonymous.

---

## Round 0.2 · Minor Revisions

Dear authors,

I have accepted the reviewers 'minor revision' decision. As you can see these are all very easy to handle, and I see no issue with you being able to get the MS back swiftly.

I look forward to receiving your revised manuscript.

Reviewer 3 ·

Basic reporting

Congratulations on a nice study of plesiosaur gastroliths and a very well-written paper. I just have some ridiculously minor comments to add that were not even enough to justify providing an annotated PDF of the manuscript.
-Maybe try substituting "body cavity" for something else as there is this connotation with soft-tissue viscera associated with this word. Up to you.
-line 103- use elasmosaurid instead of elasmosaur
-The use of "first" Wapuskanectes and "second" Wapuskanectes is hard to follow- you could say "holotype" and refer to the additional specimen by it's number.
-lines 113 & 122- check the use of "fore limb" vs "forelimb"
-line 144: "well-preserved"
-line 321: justify 900gm/l with some reference
-line 449: italicize Nichollsaura & Wapuskanectes

Experimental design

no comment

Validity of the findings

no comment

---

## Round 0.3 · accepted · Accept

Dear author,

Based on your revised manuscript, I am happy to say that your article has been accepted for publication.

The production team will contact you to take you through the proofing stages.

Congratulations, and I hope you will use PeerJ as your publication venue again in the future.